# Storytelling of Myocardial Biopsy

**DOI:** 10.3390/biology14030306

**Published:** 2025-03-18

**Authors:** Gaetano Thiene

**Affiliations:** Department of Cardiac, Thoracic, Vascular Sciences and Public Health, University of Padua, Via A. Gabelli, 86-35121 Padova, Italy; gaetano.thiene@unipd.it

**Keywords:** endomyocardial biopsy, cardiac transplantation, cardiomyopathies, ECMO, myocarditis

## Abstract

Biopsy is the piece of tissue taken from a living patient. As far as the heart, the current practice is to remove myocardium from the endocardium of the right ventricle, via a venous pathway, by a catheter with a bioptome at the extremity, able to catch endomyocardium. Inflammatory, storage, fibrotic, neoplastic tissues are the targets. With cardiac transplantation, the whole heart of the recipient is available as a big biopsy. Various pathological investigation techniques may be used: histology, immunohistochemistry electron microscopy and even molecular, the latter in search of infective organisms or gene mutations. Transplantation gives the opportunity to study the whole heart of the recipient, with the possibility to study the basic cardiac disease and even discover new morbid entities. A chance that in the past was reserved only to autopsy. Moreover, periodically performed endomyocardial biopsy allows to check the occurrence of rejection, to be treated by life saving drug therapy.

## 1. Introduction: History of Cardiac Biopsy

Biopsy means to remove a piece of or an entire organ from a living patient.

As far as the heart is concerned, needle percutaneous transthoracic biopsy was first accomplished by Casten and Marsh in 1953 [1] and by Kent et al. in 1956 [2].

Since 1954, open heart surgery with extracorporeal circulation has made it possible to perform a cardiac biopsy.

Transvenous endomyocardial biopsy (EMB) was carried out for the first time by Sakakibara and Konno in 1963 [3] Figure 1 and by Richardson in 1974 [4] Figure 2, through a bioptome at the extremity of the catheter.

## 2. Cardiac Transplantation and Cardiomyopathies

With cardiac transplantation, Christiaan Barnard in 1968 and Shumway in 1969 removed the whole heart (“cardiectomy”) and implanted a donor heart. The entire heart of a living human being was sent from the surgical theatre by the surgeon to the pathologist. It was a novel opportunity to discover and investigate new cardiac morbid entities.

A particularly dilated cardiomyopathy (DCM), the most frequent indication for a heart transplant, presented the opportunity to study the so-called “cor bovinum” in depth Figure 3.

Primary restrictive cardiomyopathy (RCM) represents the paradox of a small heart requiring a transplant Figure 4a,b, because the ventricles are unable to dilate during diastole, causing severe congestive heart failure. Since the histology shows myocardial disarray and the genetic background revealed that sarcomeric proteins are involved as in hypertrophic cardiomyopathy (HCM), this condition was named “HCM without hypertrophy”.

Also during cardiac transplantation, an amazing cardiomyopathy was discovered, named “non-compact left ventricle”, accounting for poor contractility and consisting of coarse trabeculae and deep intertrabecular fissures, with the endocardium almost reaching the epicardium Figure 5. Trabeculae observed via a 2D echo may wrongly be interpreted as mural thrombi.

Hemochromatosis is a secondary cardiomyopathy with myocardium, appearing brown to the naked eye because of myocardial storage of haemoglobin iron Figure 6. Multiple blood perfusions can be the cause.

Also, primary cardiac tumours can be diagnosed for the first time during a heart transplant.

Cardiac fibroma is often located within the ventricular septum, mimicking the asymmetric septal hypertrophy of hypertrophic cardiomyopathy, which may be wrongly diagnosed by an echo prior to heart transplantation Figure 7.

## 3. Arrhythmogenic Cardiomyopathy

Another adventure in the history of medicine was arrhythmogenic cardiomyopathy (ACM), which was discovered by Sergio Dalla Volta in 1961 [6] Figure 8a. At angiography, the right ventricle was so enlarged that the name “auricularization of the right ventricle” was employed Figure 8b. Later, with the advent of cardiac transplantation, this patient was successfully operated on in 1989. The recipient heart showed a huge dilatation of the right ventricular cavity with thin, translucent fibro-fatty wall Figure 8c,b. Oddly enough, in 1961, the “auricolarization” of the right ventricle was considered a sequela of myocardial infarction. At that time, coronary angiography did not exist yet and coronary arteries could only be investigated during an autopsy.

## 4. History of Cardiac Catheterization

A milestone in the history of EMB dates back in 1929, when the German urologist Werner Forssmann [7] Figure 9a invented cardiac catheterization, successfully accessing his own right ventricle Figure 9b with a ureteral catheter inserted into his left radial vein.

The catheter of EMB reaches the trabeculae of the ventricular septum of the right ventricle, where EMB is performed Figure 10.

## 5. The Discovery of Microscope

In 1665, Robert Hooke invented the microscope [8] to observe cells (“minute” bodies), which otherwise could not be discovered (“micrographia”) Figure 11.

## 6. Monitoring of Cardiac Transplant Rejection by Endomyocardial Biopsy

For monitoring of cardiac rejection Figure 12, in 1970, Margaret Billingham Figure 13a invented a new transvenous approach through the right jugular vein Figure 13b, which made it much easier and quicker to perform EMB weekly, after cardiac transplantation.

## 7. Clinical Indications of Endomyocardial Biopsy

Concerns regarding EMB were expressed in the early season by authoritative pathologists like Ferrans and Roberts, who questioned its utility [9] Table 1.

EMB proved to be of great help in the diagnosis of cardiac tumours. A mass found via 2D echo at the right atrioventricular grove underwent EMB with specific immunohistochemistry, which facilitated a suggested diagnosis of angiosarcoma [10] Figure 14.

Other examples, in our experience, of malignant neoplasms of the heart with diagnosis achieved by EMB immunohistochemistry include T cell atrial lymphoma Figure 15 [11] and cardiac fibrosarcoma Figure 16 [12].

EMB is also particularly useful for differential diagnosis of non-malignant masses, like Loeffler’s eosinophilic disease of the endocardium, known as obliterative restrictive cardiomyopathy Figure 17 [13].

Table 2 shows indications for EMB.

As far as myocardial inflammatory disease (myocarditis) is concerned, EMB plays a crucial role in ascertaining the histotype (lymphocytic, neutrophil, eosinophil, granulomatous non-caseous, and giant cells) Figure 18, which represents a fundamental source of information for therapeutic strategies [14,15].

## 8. The Invention of Polymerase Chain Reaction: Molecular Pathology

Lymphocytic myocarditis is usually viral. Following the invention of polymerase chain reaction (PCR) by Kary Mullis in 1983 Figure 19 [16], it is now possible to establish the type of virus by EMB [16].

Life-threatening viral myocarditis affects not only adults but also children and infants Figure 20 [17].

Accurate interpretation of EMB is currently feasible not only through histology and light microscopy (staining histologic sections, histochemistry, and immunohistochemistry) but also through electron microscopy, molecular biology with in situ hybridization, PCR, and gene sequencing. Molecular analysis is part of the diagnostic gold standard Table 3 [18].

Clinical investigation by electroanatomic mapping may be strategic for detecting scars in myocardial diseases, like in ACM, and determining where to perform EMB Figure 21.

## 9. Diagnosis of ACM Through Endomyocardial Biopsy and Infiltrative Disease as Amyloidosis

In vitro investigation of autopsy heart specimens of ACM Figure 22 established that a residual myocardium <59% is diagnostic, with specificity of 90% and sensitivity of 80% Figure 23 [19].

Diagnosis of infiltrative diseases like amyloidosis can easily be detected by EMB and using specific staining like Congo Red and Tioflavine staining Figure 24.

## 10. EMB and Electron Microscopy

Electron microscopy is particularly useful for the diagnosis of Fabry disease (lysosomes storage disease) Figure 25 and for detecting apoptosis as a mode of cardiomyocyte death Figure 26 and disruption of the desmosomes in ACM Figure 27.

## 11. Endomyocardial Biopsy and Myocarditis: The Invention of Extracorporeal Membrane Oxygenation and Its Efficacy

As far as myocarditis is concerned, the “Dallas Criteria” postulated that to achieve a diagnosis, [20,21,22] myocyte necrosis is the “condition sine qua non” in association with inflammatory infiltrate Figure 28. The observation of myocardial inflammation with interstitial oedema Figure 29 in the absence of myocardial necrosis [20] explains why extracorporeal membrane oxygenation (ECMO) Figure 30a, invented by Robert Bartlett (1939-) [22] Figure 30b, fulminant myocarditis, presenting with sudden onset of severe pump failure, resolves spontaneously because of reversible myocardial injury. ECMO may temporarily supply myocardial contractility until spontaneous recovery occurs. Ejection fraction may increase in fulminant myocarditis, with a high rate of survival Figure 31.

A left ventricular assistance device (LVAD) is another mechanical support therapy, and a source of large number of samples from myocardial specimens, available for in vivo histological and molecular analysis (biopsy).

## 12. Conclusions

EMB is the gold standard device for the diagnosis of myocarditis, cardiac rejection and infiltrative/storage disorders. The histotypes resulting from EMB can be used to inform drug therapy strategies. Molecular biology techniques should be routinely carried out as a useful complementary investigation. Sampling is fundamental (Table 4). The gold standard for diagnosis of inflammatory cardiomyopathy is a combination of histology, immunohistochemistry and molecular analysis (Table 3). EMB should be performed only if the whole “armamentarium” for pathology investigation is available.

## Figures and Tables

**Figure 1 biology-14-00306-f001:**
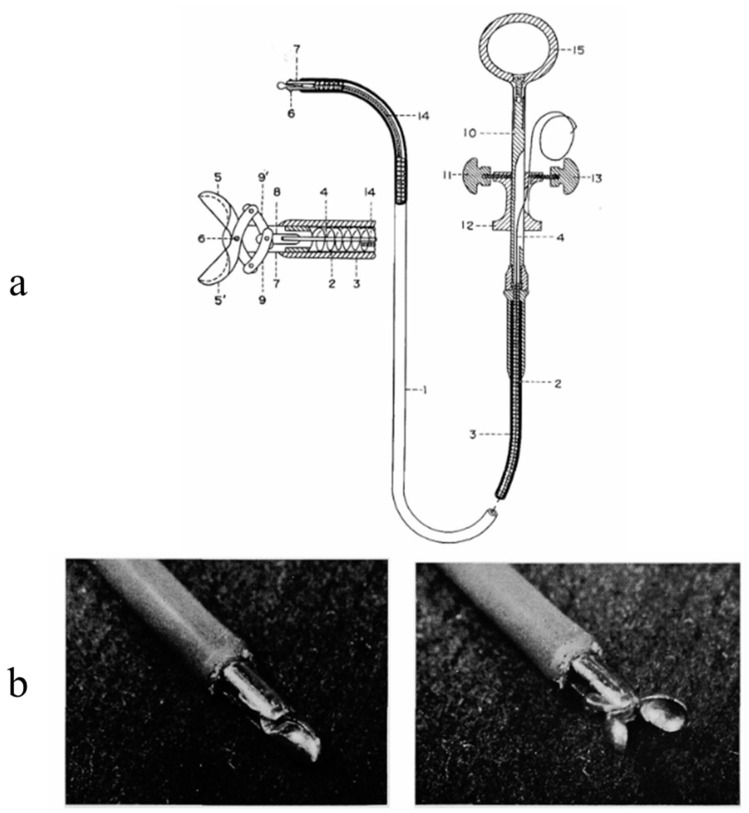
Instrument used for endomyocardial biopsy by Konno (**a**). The technique of transvenous endomyocardial biopsy (EMB), introduced in Japan by Sakakibara and Konno in 1963 [3], with the bioptome at the extremity of the catheter (**b**).

**Figure 2 biology-14-00306-f002:**
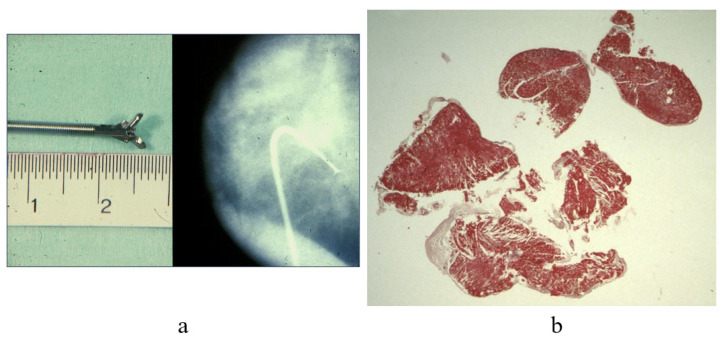
Endomyocardial biopsy (EMB) performed via a transfemoral venous approach (**a**) with endomyocardial fragments (**b**) [4].

**Figure 3 biology-14-00306-f003:**
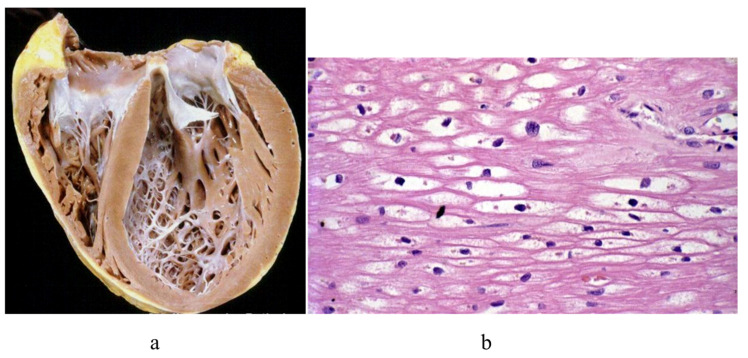
Cor bovinum and dilated cardiomyopathy in the first cardiac transplantation in Italy. (**a**) Gross view of the heart; (**b**) histology with myocytolysis performed using Hematoxylin–Eosin stain.

**Figure 4 biology-14-00306-f004:**
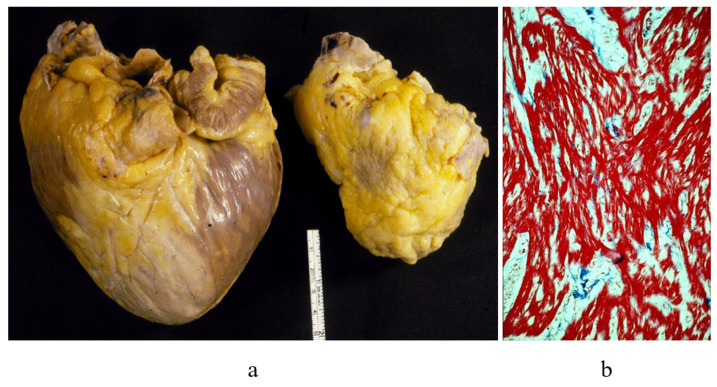
Restrictive cardiomyopathy. (**a**) Comparison between dilated vs. restrictive cardiomyopathies. (**b**) Histology of the myocardium in restrictive cardiomyopathy with disarray. Azan Mallory staining.

**Figure 5 biology-14-00306-f005:**
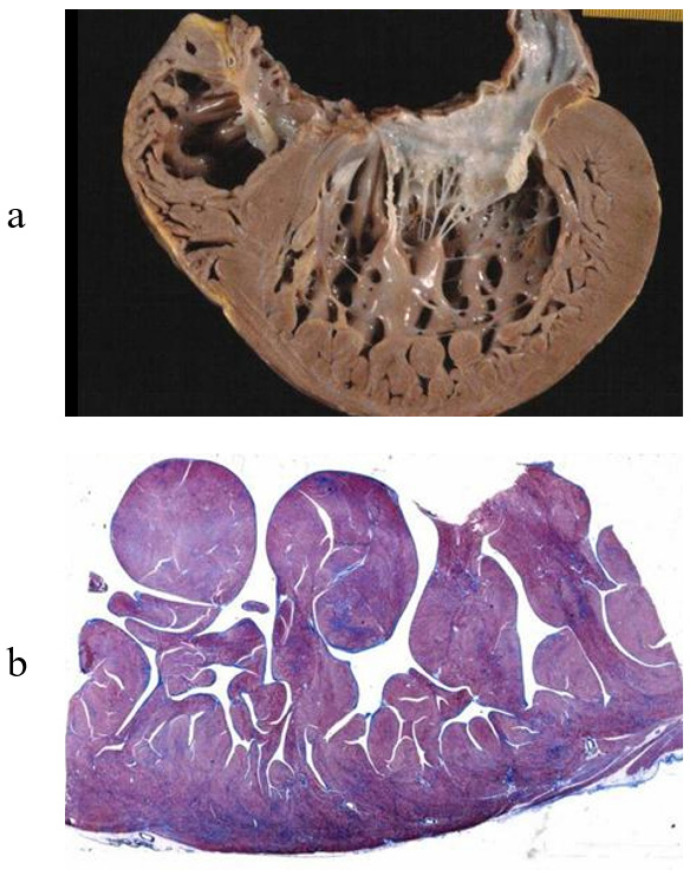
(**a**) Non-compact left ventricle; (**b**) coarse trabeculae, wrongly interpreted as thrombi by ECO.

**Figure 6 biology-14-00306-f006:**
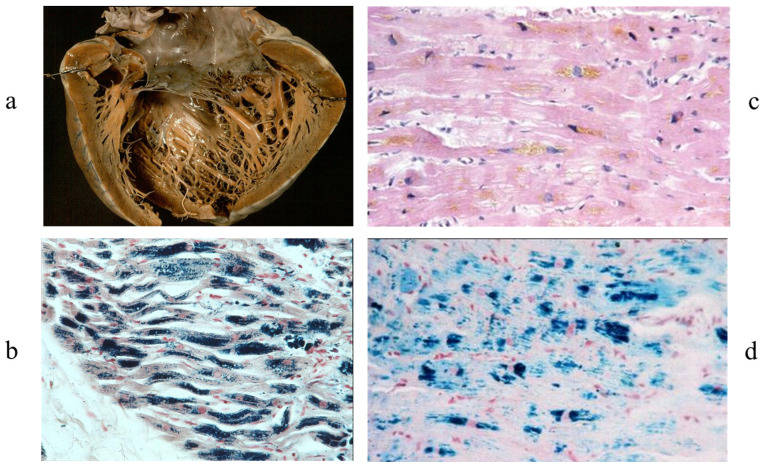
Hemochromatosis in a heart specimen. (**a**) Brown myocardium at gross examination. (**b**) Storage of iron within the cardiomyocytes. (**c**,**d**) Endomyocardial biopsy with iron storage in cardiomyocytes.

**Figure 7 biology-14-00306-f007:**
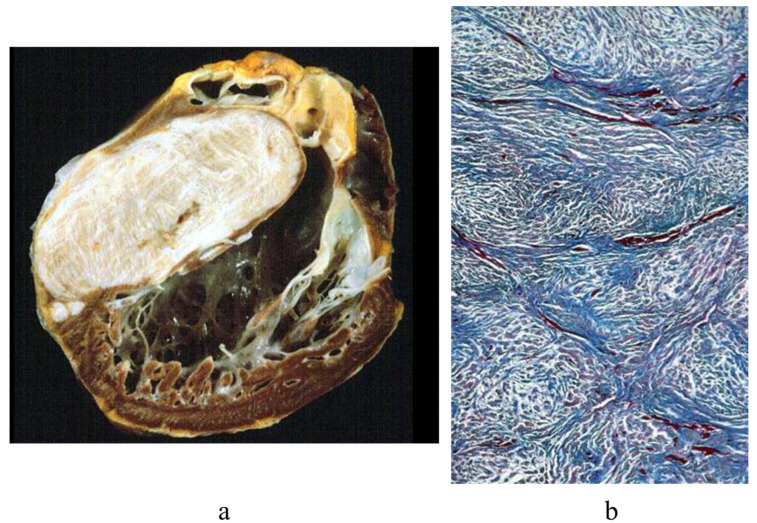
Huge cardiac fibroma. (**a**) Gross view shows involvement of the ventricular septum mimicking hypertrophic cardiomyopathy [5]. (**b**) At histology, the mass consists of collagen. Azan Mallory staining was used [5].

**Figure 8 biology-14-00306-f008:**
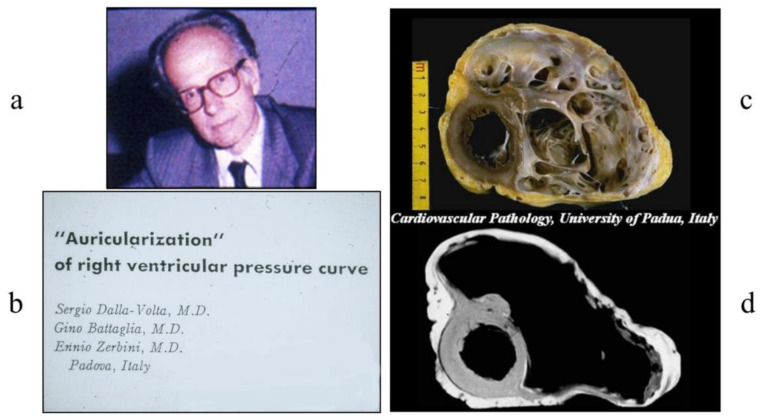
An image of a female with ACM published by Sergio Dalla Volta (**a**) in 1961 [6] (**b**), showing huge dilatation of the right ventricular cavity and paper-thin free wall (**c**,**d**). In 1996, she underwent a successful cardiac transplantation due to congestive heart failure.

**Figure 9 biology-14-00306-f009:**
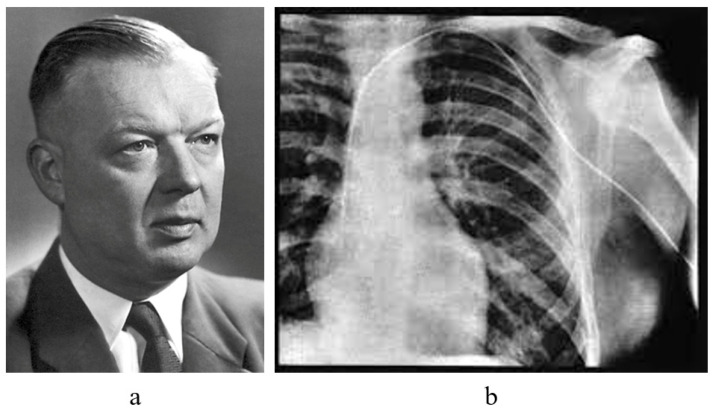
Unlike skeletal muscle, once, the myocardium could be investigated only at autopsy. With the advent of cardiac catheterization in 1929 by the German urologist Werner Forssmann (1904–1979) (**a**), it became feasible to safely reach the right ventricle in vivo (**b**) [7].

**Figure 10 biology-14-00306-f010:**
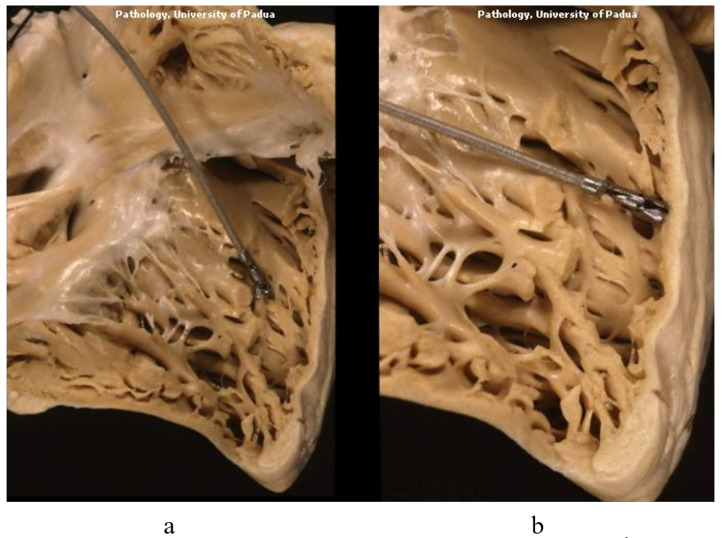
A catheter with a bioptome within the right ventricle (**a**) and septal trabeculae (**b**), where endomyocardial biopsy is performed.

**Figure 11 biology-14-00306-f011:**
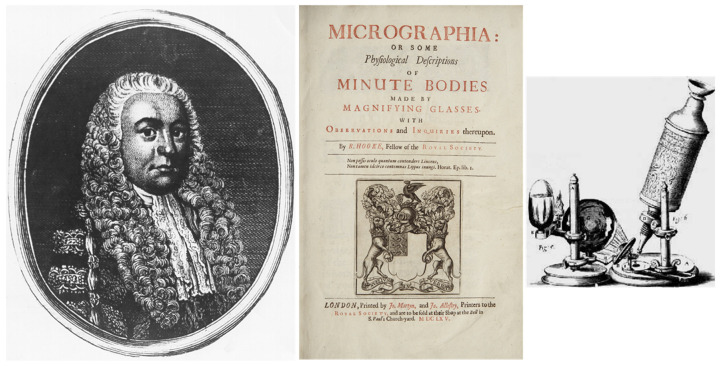
Robert Hooke (1635–1703) invented the microscope in 1665, to make micrographia of minute bodies [8].

**Figure 12 biology-14-00306-f012:**
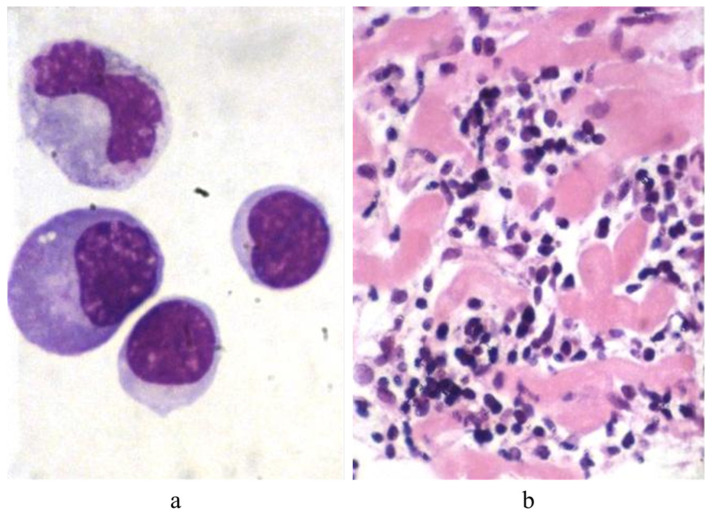
Rejection of the heart. (**a**) T lymphocytes within the blood circulation. (**b**) Myocardium infiltrated with T lymphocytes.

**Figure 13 biology-14-00306-f013:**
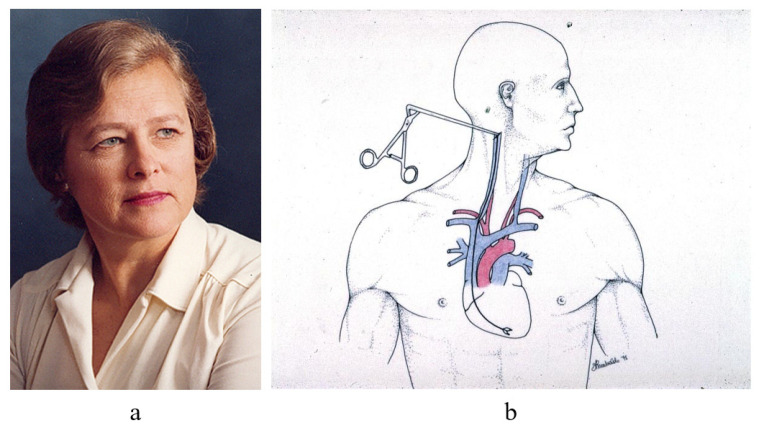
In 1970, Margaret Billingham (**a**) (1930–2009) invents EMB through the jugular vein for monitoring cardiac rejection; (**b**) a transjugular venous approach.

**Figure 14 biology-14-00306-f014:**
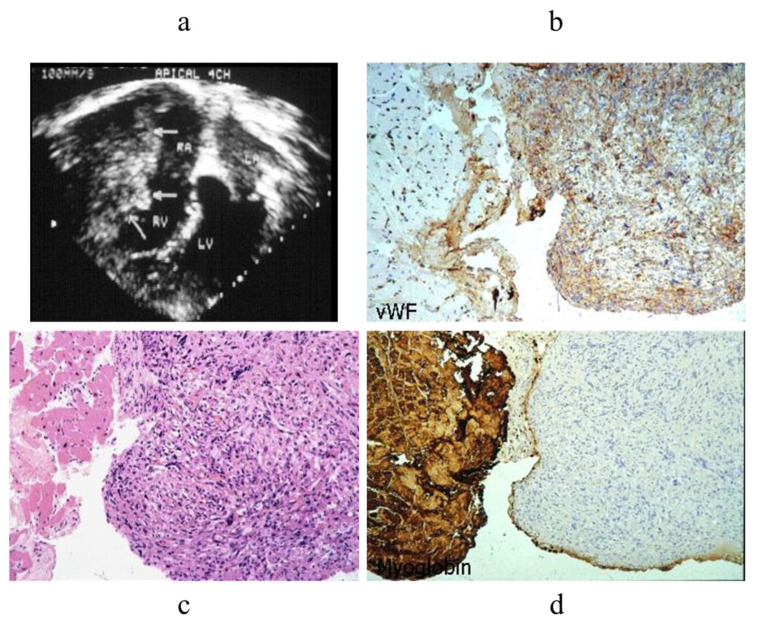
EMB of a mass at RV grove (**a**). Immunohistochemistry revealed the mass to be an angiosarcoma (**b**–**d**) [10].

**Figure 15 biology-14-00306-f015:**
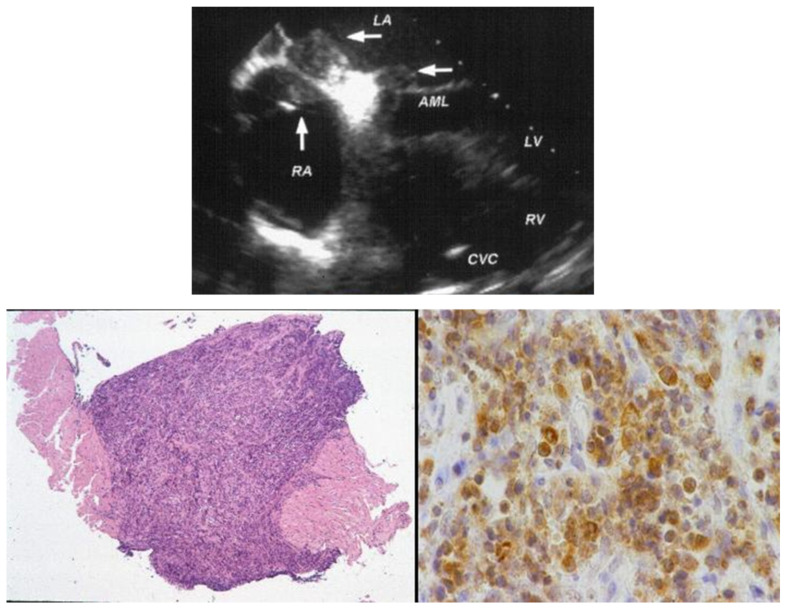
EMB of an atrial lymphoma diagnosed by immunohistochemistry [11].

**Figure 16 biology-14-00306-f016:**
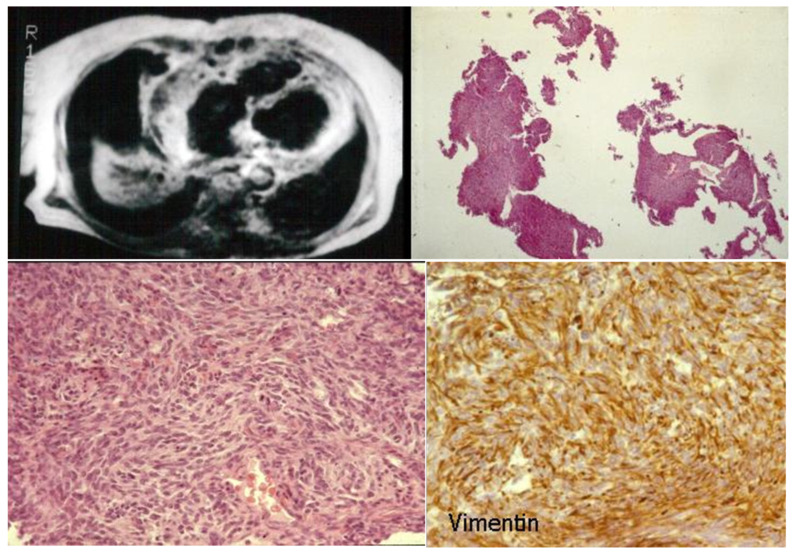
EMB of a cardiac fibrosarcoma diagnosed by immunohistochemistry [12].

**Figure 17 biology-14-00306-f017:**
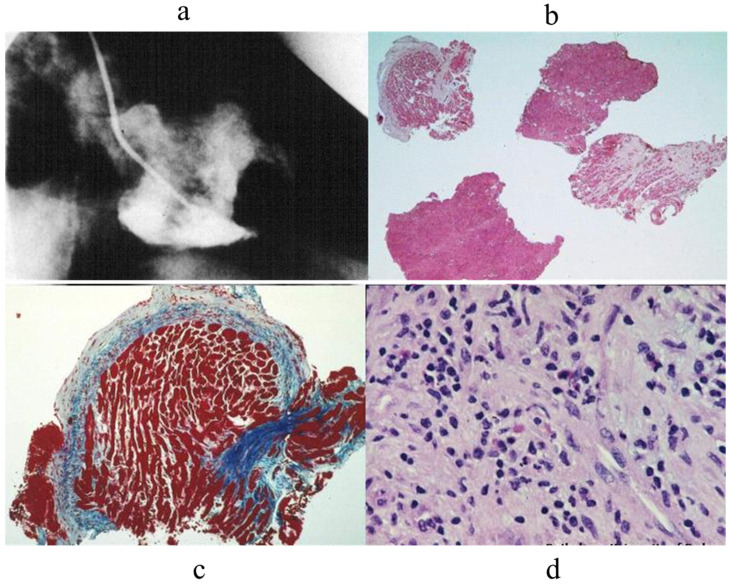
EMB in obliterative cardiomyopathy of Loeffler disease (**a**–**c**) with eosinophil infiltrates (**d**) [13].

**Figure 18 biology-14-00306-f018:**
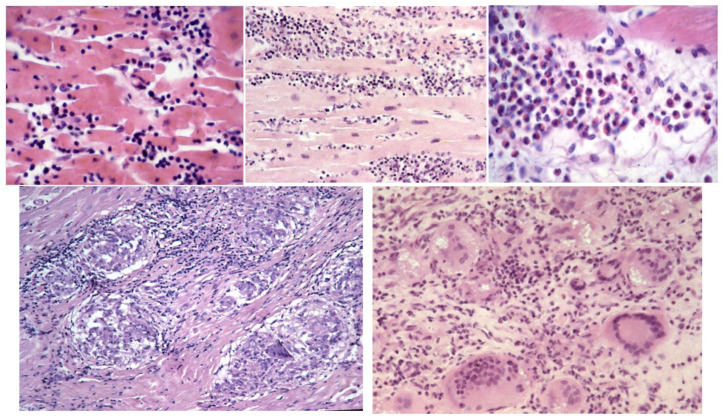
Multiple histotypes of myocarditis: lymphocyte, neutrophil, eosinophil, and giant cells [14,15].

**Figure 19 biology-14-00306-f019:**
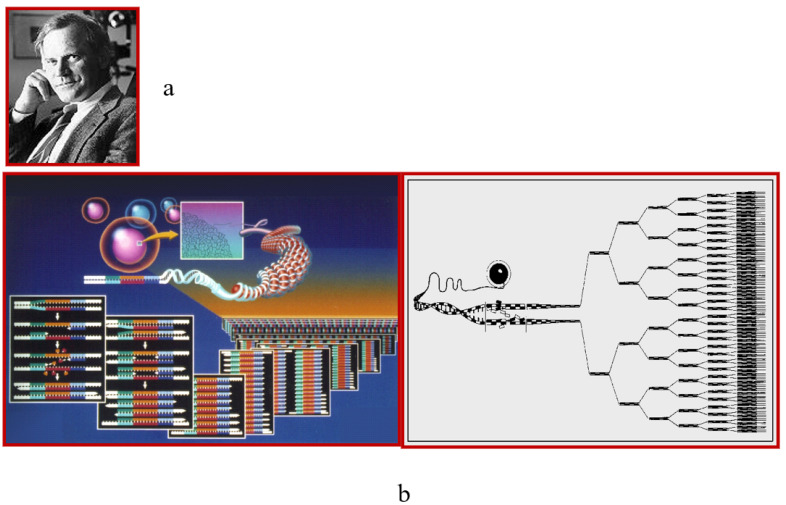
In 1983, Kary Mullis (**a**) invents the polymerase chain reaction (PCR) (**b**) [16].

**Figure 20 biology-14-00306-f020:**
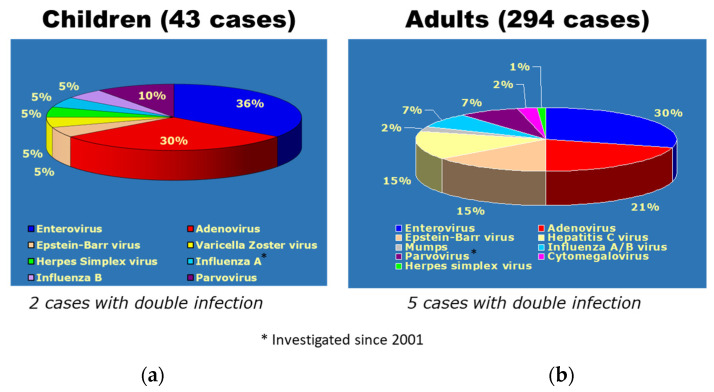
Viral PCR-proven myocarditis both in children (**a**) and adults (**b**) [17].

**Figure 21 biology-14-00306-f021:**
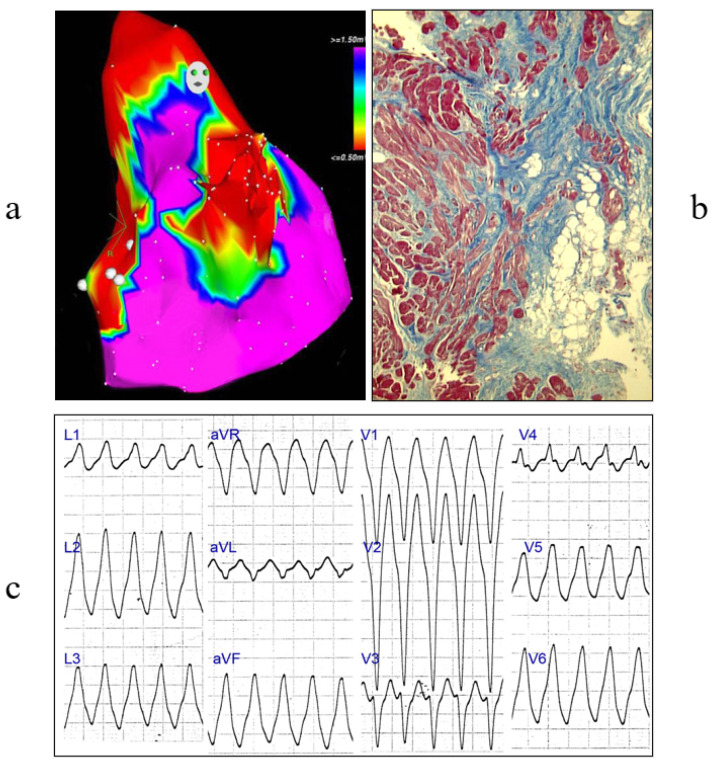
EMB in ACM. (**a**) Electroanatomic mapping with electric scar; (**b**) fibroadiposis (Azan Mallory stain); (**c**) ventricular tachycardia.

**Figure 22 biology-14-00306-f022:**
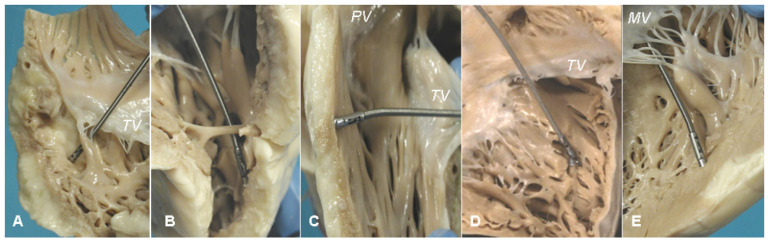
EMB in ACM with in vitro investigation of specimens at autopsy. Right ventricle with fibroadiposis of the free wall (**A**,**B**). The catheter with the bioptome simulating endomyocardial biopsy (**C**). Other pictures of the right ventricle and the catheter with the bioptome (**D**,**E**).

**Figure 23 biology-14-00306-f023:**
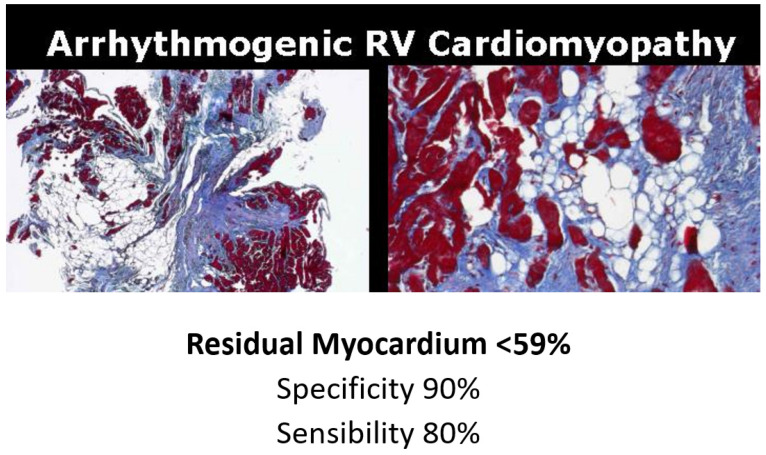
Specificity (90%) and sensitivity (80%) of the residual myocardium (<59%) in the diagnosis of arrhythmogenic cardiomyopathy [19].

**Figure 24 biology-14-00306-f024:**
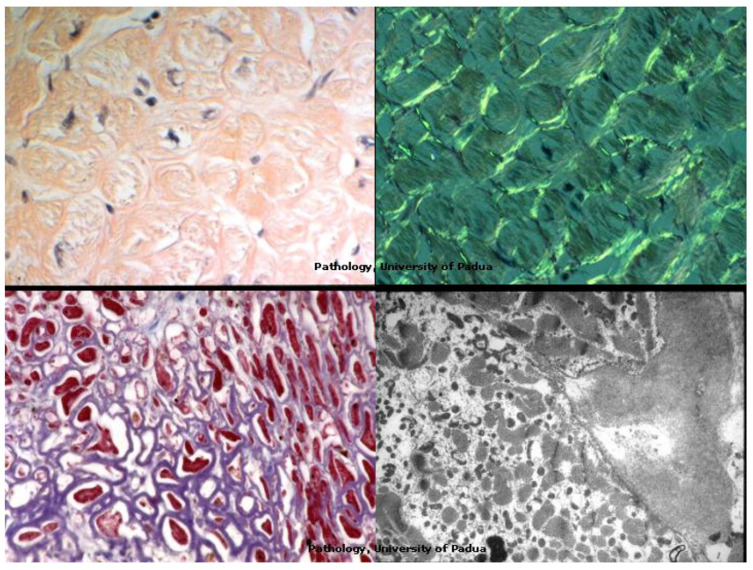
Diagnosis of amyloidosis by EMB as infiltrative diseases of the myocardial interstitium with specific stems.

**Figure 25 biology-14-00306-f025:**
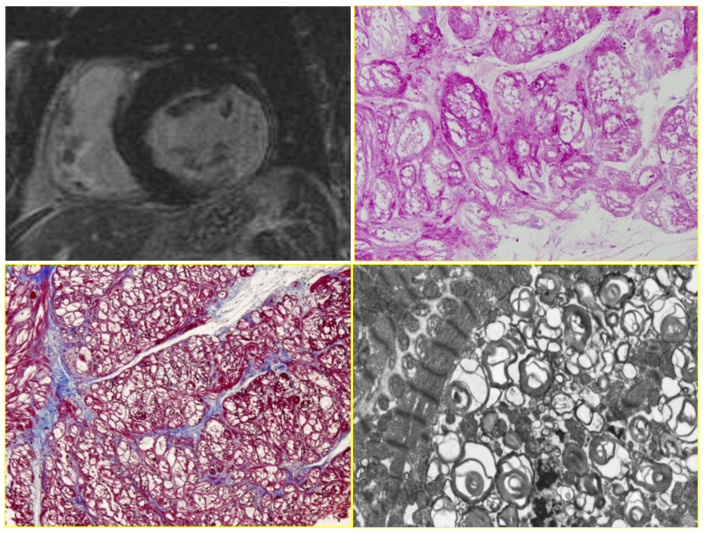
Fabry disease: a secondary hypertrophic cardiomyopathy diagnosed by electron microscopy.

**Figure 26 biology-14-00306-f026:**
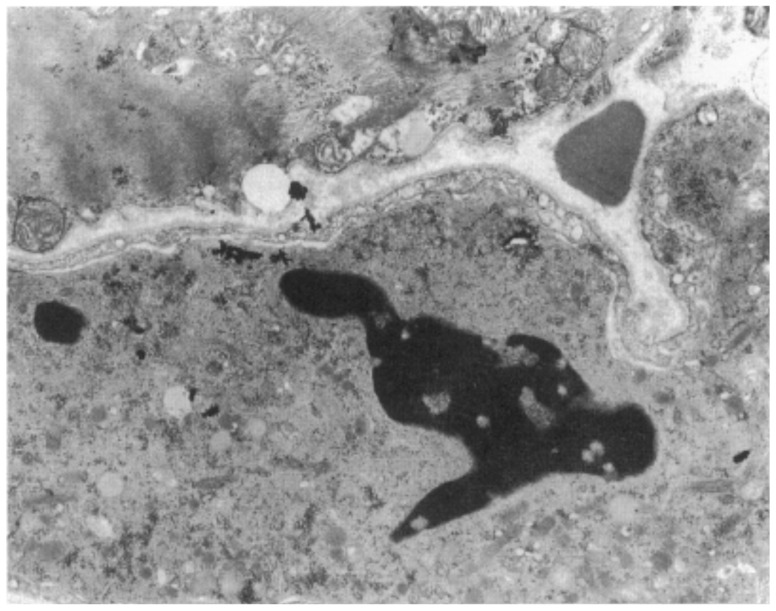
Electron microscopy of myocardial apoptosis in ACM. Original magnification, ×14,500.

**Figure 27 biology-14-00306-f027:**
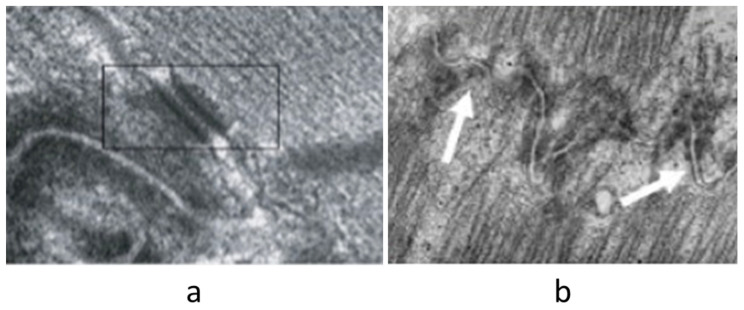
Electron microscopy in ACM. Transmission electron microscopy shows a disrupted intercalated disc (**b**) compared to normal desmosome (**a**). Original magnification, (**a**) ×30,000; (**b**) ×50,000. Rectangular block includes a normal desmosome (**a**), whereas the arrow indicates disrupted desmosomes in AC (**b**).

**Figure 28 biology-14-00306-f028:**
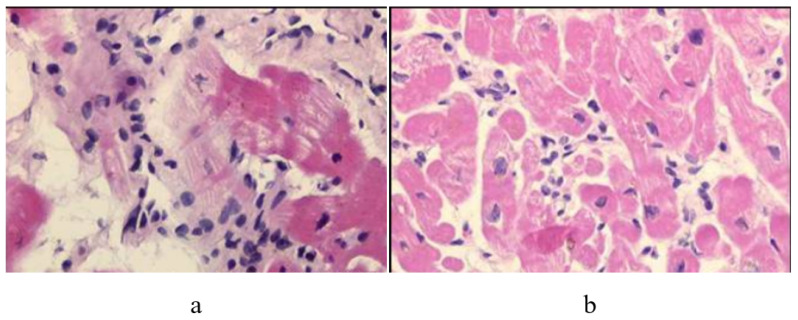
Dallas Criteria. Active myocarditis was possible to detect in vivo by EMB and diagnostic histologic criteria were put forward in Dallas by the Society for Cardiovascular Pathology in 1985, based upon microscopic observation of inflammatory infiltrates associated with myocardial death (“Dallas Criteria”). (**a**) Active myocarditis. Inflammatory cellular infiltrate with evidence of myocyte necrosis. (**b**) Borderline myocarditis. Inflammatory cellular infiltration without evidence of myocyte injury. Hematoxylin–Eosin staining was used [20].

**Figure 29 biology-14-00306-f029:**
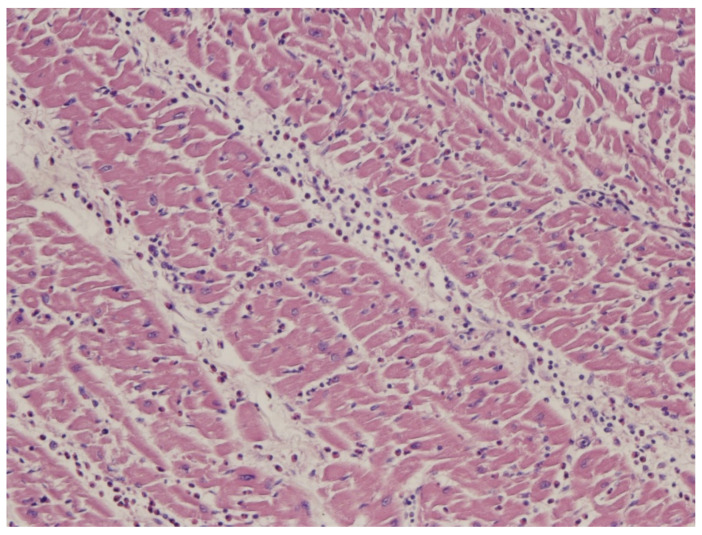
Lymphocytic myocarditis with inflammatory infiltrate and the oedematous interstitium, without cardiomyocytes necrosis.

**Figure 30 biology-14-00306-f030:**
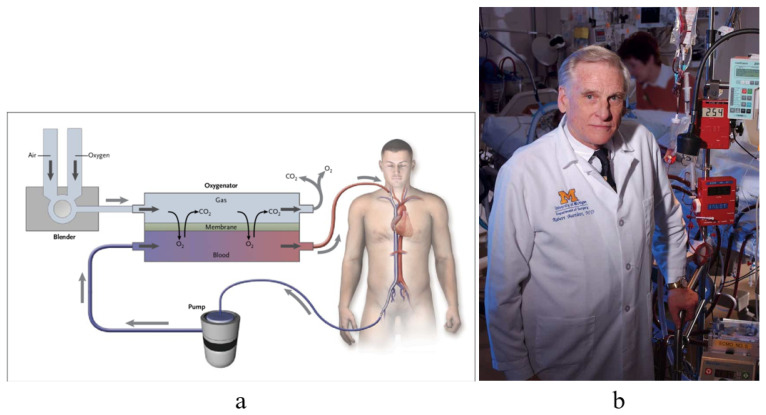
Robert Bartlett (1939-) (**b**) developed the tool (**a**) of extracorporeal membrane oxygenation (ECMO) [22].

**Figure 31 biology-14-00306-f031:**
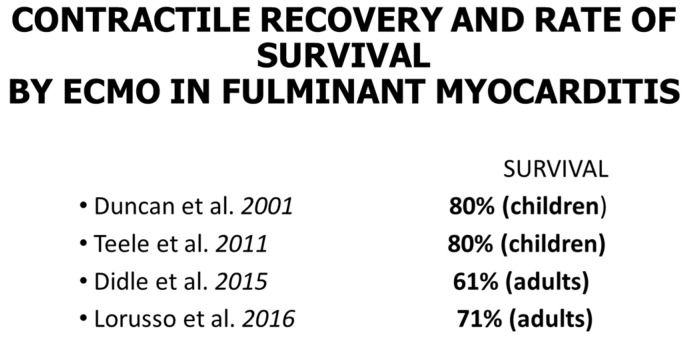
Contractile recovery and rate of survival achieved when employing ECMO in fulminant myocarditis (both in children and adults) [23,24,25,26].

**Table 1 biology-14-00306-t001:** Questioning the utility of EMB [9].

How representative is a biopsy segment of the state of the whole heart?
Is endomyocardial biopsy useful enough to balance the risks?
Does endomyocardial biopsy play a role in clinical decision making?

**Table 2 biology-14-00306-t002:** Indications for EMB.

**Cardiac transplantation monitoring**
**Non-ischemic heart failure**
**Arrhythmias**
**Chest pain with normal coronary arteries**
**Cardiac masses**

**Table 3 biology-14-00306-t003:** Inflammatory cardiomyopathy. Endomyocardial biopsy [18].

Molecular Analysis	Histology	Immunohistochemistry
↘	↓	↙
	**Gold Standard**	

**Table 4 biology-14-00306-t004:** EMB sampling.

**At least 4–5 samples, 1–1.5 mm size**
**From different topographic sites**
**Investigation: 3–4 samples for histology and immunohistochemistry (10% buffered formalin)**
**Serial histology sections**
**1 sample for electron microscopy (fixed in 2% glutaraldehyde) in selected cases**
**1–2 samples for molecular biopsy (frozen −80°)**

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
