# Peer review of "Storytelling of Myocardial Biopsy"

_biology, 2025, doi:10.3390/biology14030306_

Round 1

Reviewer 1 Report

Comments and Suggestions for Authors

Dear Professor Gaetano Thiene!
I would be honoured to review your paper, which has allowed me to get to know it already at this stage. 

As a reviewer, I can only express my congratulations and wish the article to be published in its present form as soon as possible.

Comments on the Quality of English Language

See above, please

Author Response

I deeply thank you for the congratulations and the wish the article to published in the present form as soon as possible. As far as the quality of the English Language, I leave to the Publisher to employ its service.

Reviewer 2 Report

Comments and Suggestions for Authors

This is an important review that increases our knowledge

Of the past and present of the endomyocardial biopsy

The paper is written by a scientific autority in this field

I belive that all of us and in particular the young readers

need this type of papers

Author Response

Quality of English Language

According to this reviewer, the quality of the English language is fine and does not require any improvement.

Comments and Suggestions for Authors

I welcome the compliments “This is an important review that increases our knowledge […] and in particular the young readers need this type of papers”.

Reviewer 3 Report

Comments and Suggestions for Authors

Dear Author, Dear  Editor,

I have read with a great interest the “Storytelling of myocardial biopsy”.

This article presents the evolutional history and progress done during almost half of the century, recalls the names of big scientists which contributed to the progress in the instrumentation of myocardial biopsy techniques, access possibilities and evolving clinical significance of this technique.

Especially interesting for me and for younger adepts of medicine are the historical aspects of this technique presented in this article. This part of the article is illustrated by multiple remarkably interesting photographs as I understand taken from the private collection of the Author or reproduced with permission from various sources.

The presentation of the history of this myocardial biopsy techniques and evolving clinical aspects of its use is e real strength of this article.

My minor critical remarks are as follows.

  1. The newest techniques, which were developed recently like molecular microscope in transplant rejection detection, PCR viral specimen detection, are only signaled in this article, however, were not presented in detail.
  2. Author mention ECMO as a bridge to recovery therapy in myocarditis- it will be also important to mention other available modes of mechanical support therapy like LVAD implantation. Additionally, LVAD seems to be worth mentioning in the scope of this article due to the fact that during LVAD implantation a large sample of myocardial specimen is available for analyses -one of the largest specimens available form the left ventricle- which can be very important for the diagnosis.
  3. Small native speaker linguistic changes will be welcome
Comments on the Quality of English Language

Dear Author, Dear  Editor,

  1. Small native speaker linguistic changes will be welcome

Author Response

Quality of English Language

I leave the Publisher to employ their service to ameliorate the English.

Comments and Suggestions for Authors

I deeply express my gratitude for the nice comments. They are greatly rewarding.

Minor remarks

  • I believe that PCR viral detection in endomyocardial biopsy has been sufficiently treated.
  • LVAD was added as available mechanical device. I underlined that it may be a source of myocardium for histological and molecular analysis.